# Comparing the Results of Vitrectomy and Sclerectomy in a Patient with Nanophthalmic Uveal Effusion Syndrome

**DOI:** 10.3390/medicina57020120

**Published:** 2021-01-29

**Authors:** Ivajlo Popov, Veronika Popova, Vladimir Krasnik

**Affiliations:** 1Department of Ophthalmology, Faculty of Medicine, Comenius University, 82101 Bratislava, Slovakia; ivajlo.popov@gmail.com; 2Department of Pediatric Ophthalmology, Faculty of Medicine, Comenius University, The National Institute of Children’s Diseases, 83101 Bratislava, Slovakia; veronika.labuzova@gmail.com

**Keywords:** nanophthalmos, uveal effusion syndrome, serous retinal detachment, sclerectomy

## Abstract

Nanophthalmic uveal effusion syndrome (UES) is an extremely rare idiopathic disease characterized by a short axial length of the eye, extremely thick sclera and choroid. These structural changes can lead to spontaneous serous detachment of the retina and peripheral choroid. There are many other causes of UES such as trauma, inflammation, cataract surgery, glaucoma, or retinal detachment. UES is classified into three types. All are characterized by a relapsing-remitting clinical course. The loss of visual acuity ranges from mild to very severe, depending on macular involvement. Changes of the retinal pigment epithelium develop secondary after long-standing choroidal effusion and retinal detachment. Subretinal exudates could be seen and mistakenly diagnosed as chorioretinitis. UES can be very difficult to treat. The most commonly used treatment is surgery involving the creation of surgical sclerostomies (scleral window surgery) or partial thickness sclerectomies to support transscleral drainage. In our case, we present a bilateral nanophthalmic UES, which was misdiagnosed as bilateral ocular Vogt−Koyanagi−Harada disease. We documented the course of the disease and the results of the different surgical approaches in both eyes. A pars plana vitrectomy was performed in the right eye and a sclerectomy with sclerostomies in the left eye. In the left eye, even long lasting loss of visual acuity due to a serous retinal detachment was partially reversed.

## 1. Introduction

Uveal effusion syndrome (UES) is an extremely rare idiopathic disease characterized by idiopathic spontaneous serous detachment of the retina and peripheral choroid. There are many causes of UES such as trauma, inflammation, cataract surgery, glaucoma, or retinal detachment [1]. Uveal effusion syndrome was first reported in 1963 by Schepens and Brockhurst in 17 patients with choroidal detachment often with retinal detachment [2]. There is no known cause of UES, but many theories have been postulated. These include vortex vein compression, increased choroidal permeability, abnormal scleral collagen or decreased scleral permeability [3,4,5,6]. UES is classified into three types [7]. Type 1 is caused by nanophthalmic eye, where the average axial length of the eye is 16 mm, the eye is hypermetropic (average +16D) and the sclera is significantly thickened. Type 2 and 3 UES are idiopathic and the axial length is normal. Type 2, however, demonstrates abnormal sclera with disorganization of collagen fiber bundles. Type 3 has normal histology of the sclera. Both sexes are affected equally in nanophthalmic UES but idiopathic UES show a strong male preponderance [1].

UES can be very difficult to treat. The most commonly used treatment is surgery involving the creation of surgical sclerostomies (scleral window surgery) or partial thickness sclerectomies to support transscleral drainage [1,8,9]. In some cases, steroids or NSAID may be effective [10,11].

In our case, we present a bilateral nanophthalmic UES misdiagnosed as bilateral ocular Vogt−Koyanagi−Harada disease (VKHD). We documented the results of different surgical approaches on both eyes with visual recovery after prolonged retinal detachment.

## 2. Case Presentation

In 2015, a 44-year old male was sent to our clinic by a regional ophthalmologist with the diagnosis of bilateral retinal detachments. The patient described gradual worsening of vision in both eyes from 2012. In the year before the visit, the deterioration became more rapid. In 2010, he was diagnosed with closed angle glaucoma, neodymium-doped yttrium aluminum garnet laser (Nd:YAG) iridotomies were performed bilaterally in 2012, and a cataract surgery with intraocular lens implantation in the bag was performed on the left eye in 2013. No significant posterior capsule opacifications were present at the presentation. Right eye was left phakic. Amblyopia was not noted in the available documentation.

The patient was followed up for bilateral nanophthalmos by a regional ophthalmologist from his childhood. His mother reported that he had very small eyes from birth, and that he had worn hypermetropic correction since he was one year old. No family history of nanophthalmos was reported. At the presentation, the patient’s glasses correction was +14D on the right eye and +11.5D on the left eye. Automated refractometry was not possible to obtain as visual fixation was poor.

On admission, best corrected visual acuity (BCVA) was 10/200 in the right eye and 20/200 in the left eye. Ophthalmological examination of both eyes showed hyperemic conjunctival vessel, shallow anterior chamber, and detached retinas in the inferior periphery (almost to the vascular arcade on the right eye) (Figure 1).

Indirect ophthalmoscopy revealed yellowish exudates under the retina at the level of the choroid. Short axial length (right eye—14.35 mm, left eye—14.29 mm) and thickened choroid (2.68 mm—right eye, 2.56 mm—left eye) with inferior retinal detachments were diagnosed on the ocular ultrasound B-scan (Figure 2).

Fluorescein angiography showed hyperfluorescent spots with pinpoint leakage at the early stages on both eyes (Figure 3).

Hyperfluorescent spots with pinpoint leakage from early stages are shown for both eyes. The blood tests (serology, blood count, biochemistry and coagulation) were negative. Further examinations, such as chest X-rays and USG of the abdomen, were performed. All test results were negative. The magnetic resonance imagingof the brain and orbits described a nanophthalmia, without further pathological signs.

Due to the choroidal exudates and hyperfluorescent spots on the retinal angiography, probable Vogt−Koyanagi−Harada disease (VKHD) was diagnosed. No other symptoms were present. He was treated with pulses of intravenous corticosteroids over the next few days and showed subtle improvement. Retinal detachments were still present. On discharge, oral prednisolone with slow tapering was prescribed. Specific antigens (HLA-DR4, -DR1, -DQ4, -Dw5) were sent for analysis; only HLA-DR1 was positive.

From 2015 to 2017, the patient’s BCVA undulated from 20/200 to counting fingers on both eyes. Intravenous corticosteroid pulses were administered when vision worsened. Retinal detachments were still present with periodic enlargements of the subretinal fluid. In 2017 visual acuity worsened bilaterally despite corticosteroid pulses (BCVA oculus dexter (OD): counting fingers, oculus sinister (OS): hand movement). Retinal detachment enlarged. In the left eye, bullous retinal detachment developed with involvement of the macula (Figure 4).

The patient was put on an immunosuppressive therapy with cyclosporine and was scheduled for surgery on the better right eye. The surgery was postponed because the patient was admitted to a center specializing in uveitis treatment. They started biological treatment with Adalimumab.

In April 2018, after ending biological treatment, BCVA was unchanged and right eye surgery (23G pars plana vitrectomy with silicone oil 5000 cs tamponade) was performed. One month after surgery BCVA OD improved to 10/200. Central retina was attached on the right eye. From the arcades, flat retinal detachment was present. Left eye worsened with BCVA only light perception, and the retina was completely detached and adhered to the posterior surface of the intraocular lens (Figure 5).

Attention was focused on preserving vision on the better right eye. No action was decided to be taken on left eye. In February 2019, BCVA OD worsened to hand movement. Central retina stayed attached. Second pars plana vitrectomy in the right eye with lensectomy and silicone oil 5000 cs exchange was performed due to open iatrogenic retinal break from the first surgery and cataract progression. Silicone oil was left as a permanent tamponade. BCVA OD improved to 10/200.

Due to the poor response to corticosteroids and immunosuppressive treatment, the diagnosis of Vogt−Koyanagi−Harada Syndrome was re-evaluated and a new diagnosis of nanophthalmic uveal effusion syndrome was set. Both eyes showed no functional or anatomical changes within one year. Surgery to increase scleral permeability was performed on the left eye in May 2019. Gass technique was applied [12]. In each quadrant of the globe, a half thickness scleral window of 5 × 7 mm was excised at a distance of 8 mm from the limbus. In the center of the sclerectomy, a small sclerostomy to the choroid was excised with a blade and left open. The subretinal fluid was drained through the sclerostomy. Retina fell back towards the posterior pole. Intraocular lens fell back with the retina to the vitreous cavity. After one month the retina was completely attached but the BCVA of the left eye was unchanged. Three months after surgery, the patient was very satisfied as he started to see silhouettes of objects with his left eye. BCVA of the left eye improved to the hand movement. BCVA of both eyes remained stable after one year and patient was satisfied with the stable vision. The retina of the left eye remained attached. Immunosuppressive therapy was discontinued. Patient follow-up continued.

## 3. Discussion

Uveal effusion syndrome is a very rare disease. Nanophthalmic UES is even rarer, since the majority of published cases are idiopathic UES [1]. Nanophthalmos is a disorder characterized by arrested growth of the eye during the embryonic stage [13]. Unlike microphthalmia, nanophthalmia is not associated with developmental defects of the fetal fissure, it is always bilateral with proportionally smaller ocular structures except for the crystalline lens [1]. The disorder is usually sporadic, as in our case, but familiar occurrence has been reported [14]. Patients with UES are typically middle-aged men. UES may be difficult to diagnose as it may mimic many entities. Shields in his study on 104 eyes reported a mean age of 70 years [10]. However, the study specifically evaluated Type 3 UES with normal sclera using ultrasound or MRI. Most of the cases were male (64%) with unilateral findings (87%). Many of their cases were firstly misdiagnosed as choroidal melanoma (47%), choroidal metastasis (3%), choroidal tumor nonspecified (34%) and only 16% was diagnosed correctly as UES [10]. In type 1 UES, 65% of the cases eventually become bilateral, which is the case with our patient [11].

Typical findings in UES are uveal effusion and serous retinal detachment. Both were presented in our case. A bilateral serous retinal detachment is a rare condition although it may occur in some other diagnosis. Multifocal chorioretinitis as result of sarcoidosis, tuberculosis, lymphoma, leukemia, syphilis or other diseases could have similar appearance. In our patient, chest x-ray, serology, Mantoux, blood count and other blood parameters were within the normal range, which makes the above-mentioned diagnoses unlikely. Bilateral serous retinal detachment has been reported in hypertensive retinopathy [15]; in our case, however, the blood pressure was normal.

Our patient presented with a bilateral retinal detachment with exudates under the retina and in the choroid. Sympathetic ophthalmia could mimic similar signs, therefore an ocular trauma must be ruled out. No history of ocular trauma was present in our case. The only surgery was cataract surgery, which could be the trigger for sympathetic ophthalmia, although this was unlikely since the surgery was without any complications. Cataract surgery could be the predisposing factor for uveal effusion [16], but surgery was performed only in the right eye, and UES were presented bilaterally. This makes cataract surgery an unlikely triggering mechanism.

All types of UES are characterized by a relapsing-remitting clinical course. The loss of visual acuity ranges from mild to very severe, depending on macular involvement. Changes in the retinal pigment epithelium develop secondary to long-standing choroidal effusion and retinal detachment. Subretinal exudates could be seen and mistakenly diagnosed as chorioretinitis (often as VKHD which may have similar characteristics) or toxoplasmosis [1].

American Uveitis Society defines a complete form of VKHD as a nontraumatic bilateral panuveitis, associated with integumentary, neurological/auditory signs [16]. Probable (ocular) VKHD is characterized by ocular manifestations but no extraocular manifestations and is reported in 45% of patients with acute disease and up to 58% of patients with chronic disease [17]. Ocular manifestation of the acute phase could be characterized by bilateral thickening of the choroid, serous retinal detachments and inflammatory Dalen−Fuchs nodules in the posterior pole between Bruch membrane and the retinal pigment epithelium [18]. These signs were all present in our case. No anterior chamber or vitreous inflammation were present. Vitreous inflammation is not always present within the first two weeks from the onset of symptoms [19]. Diagnosis of VKH was supported by positive HLA-DR1. However, this could be misleading as sub-allele analysis was not performed. Sub-alleles HLA-DRB1*0404, 0405, 0410 are associated with VKHD but LA-DRB1*0401, 0604 are considered protective [20,21,22].

After the new diagnosis was settled, surgical treatment was performed. Many surgical techniques were developed to address some of the theoretical causes of UES, such as vortex vein decompression or thick impermeable sclera. In the nanophthalmic eye, Brochhurst suggested sclerectomies with vortex vein decompression [8]. Gass modified this approach by using four sclerectomies including a sclerostomy enlarged with a scleral punch [12]. Casswell successfully treated an 11-year-old boy with nanophthalmos using only four partial thickness sclerectomies [9], which provides further evidence that a scleral abnormality could the underlying cause of nanophthalmic uveal effusion syndrome. Mansour enlarged the scleral window circumferentially behind the rectus muscles to over 3 and 1/4 quadrants [11]. The role of vitrectomy in nanophthalmic UES is still unclear. Successful treatment with vitrectomy was observed in Claeys study [16], although none of their patients had nanophthalmic UES. In hypermetropic eyes, they recommend the use of a vitrectomy in conjunction with superotemporal sclerectomy. Pars plana vitrectomy with silicone oil tamponade was performed in nanophthalmic UES in the Takahide study [23], although authors recommend performing sclerostomy first. The results of our case are in agreement with those recommendations.

In our case, we used the Gass technique with anatomical success. This confirms the theory of scleral impermeability. Increasing the scleral permeability by this surgical procedure enabled absorption of the subretinal fluid and kept retina attachment.

In conclusion, UES is an extremely rare disease with only 101 publications searchable in Pubmed, most of which describe idiopathic UES. Our case demonstrated outcomes of two different surgical procedures in different eyes in the same patient. Both procedures led to stabilization of best corrected visual acuity, but vitrectomy carries greater risk of complications with less favorable visual and anatomical outcomes. Pars plana vitrectomy in nanophthalmic UES should be performed only in cases with poor response to scleral surgery. Sclerectomy is the method of choice in nanophthalmic UES because it is the more effective and less risky procedure. Probably, the earlier performed sclerectomies would lead to better final BCVA but even a long-lasting loss in visual acuity due to a serous retinal detachment was partially reversed.

## Figures and Tables

**Figure 1 medicina-57-00120-f001:**
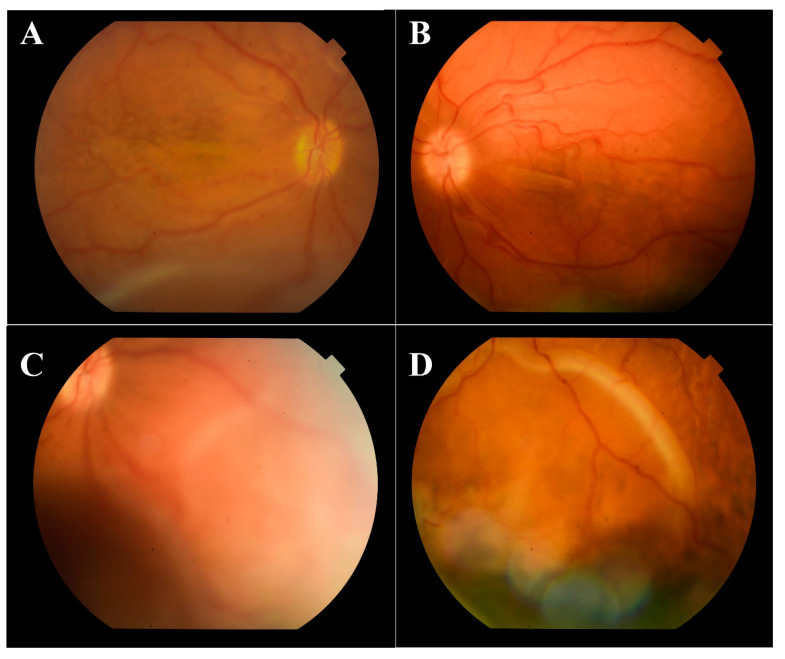
Color fundus photography of the macular region (**A**—right eye, **B**—left eye) with retinal detachment almost to the inferior temporal arcade on the right eye, and of the inferior periphery with detached retinas (**C**—right eye, **D**—left eye).

**Figure 2 medicina-57-00120-f002:**
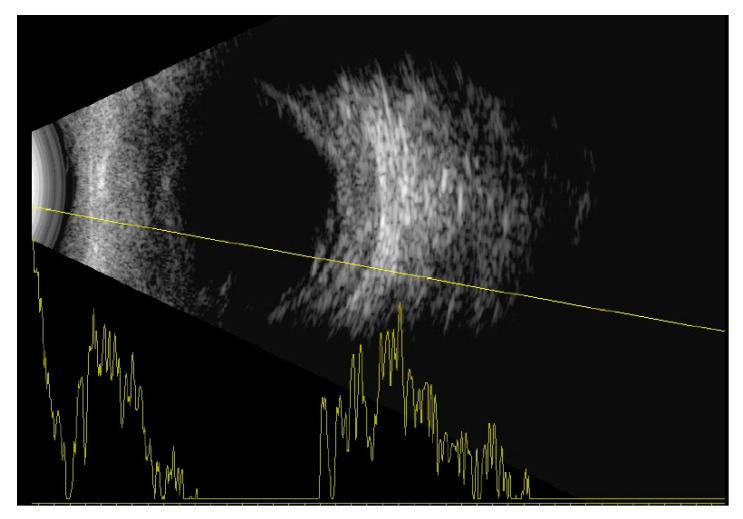
B-scan of the right eye with thickened choroid and inferior retinal detachment.

**Figure 3 medicina-57-00120-f003:**
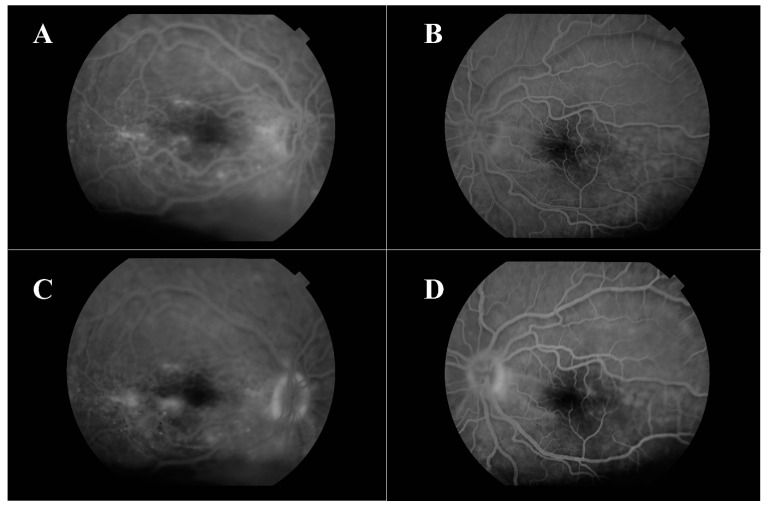
Fluorescein angiography—early stages of right (**A**) and left (**B**) eye, late stages of right (**C**) and left (**D**) eye.

**Figure 4 medicina-57-00120-f004:**
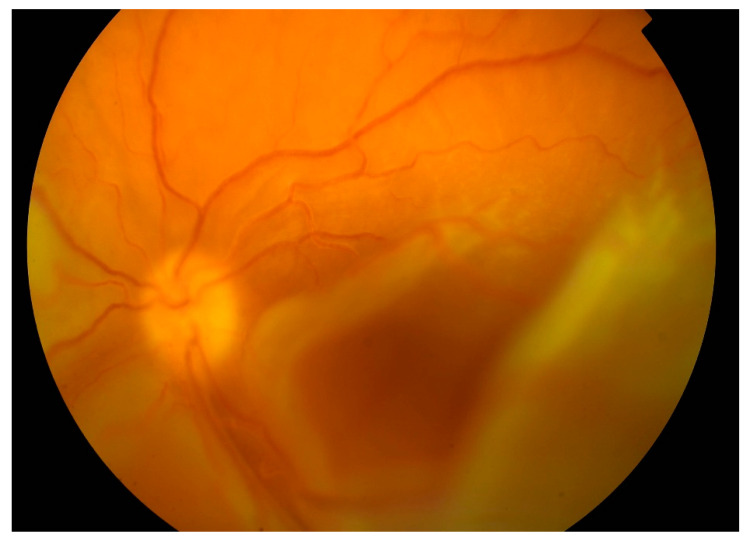
Color fundus photography of the detached retina with macular involvement on the left eye.

**Figure 5 medicina-57-00120-f005:**
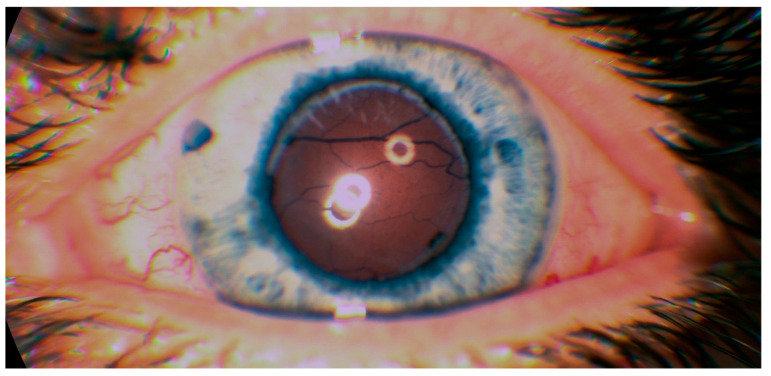
Touch of the detached retina to the posterior surface of the intraocular lens on the left eye.

## Data Availability

The data presented in this study are available on request from the corresponding author. The data are not publicly available due to data protection policies.

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
