# Peer review of "Comparing the Results of Vitrectomy and Sclerectomy in a Patient with Nanophthalmic Uveal Effusion Syndrome"

_medicina, 2021, doi:10.3390/medicina57020120_

Round 1
Reviewer 1 Report
Good description of the rare case. After long term correct diagnosis was established and proper treatment with reasonable outcome achieved.
Author Response
Dear Reviewer 1,
Thank you very much for your kind review.
We hope you will consider our manuscript suitable for publication in journal Medicina.
Thank you for your kind response
Best regards
Reviewer 2 Report
The case presentation should be improved in terms of events order from childhood to the first visit done by authors. There are some not necessary repetition (the results of fluorescein angiography).
This is an interesting case report, but unfortunately it does not add anything new to the already known UES features and treatment best option.
English needs extensive revision.
Author Response
Dear Reviewer 2,
Thank you for your review and your notes.
We tried to correct our manuscript according to your suggestions:
- events order reorganized
- English revision was done by professional proofreading agency: https://cityhillsproofreading.com/
We hope you will be satisfied with our revisions and you will consider our manuscript suitable for publication in journal Medicina.
Thank you for your kind response
Best regards

Reviewer 3 Report
- no information about potential amblopya in the history (highly hyperopic, anisometric eyes)
- no information about state of the lens after catgaract suregey in the left eye (pseudophacic?/aphacic?). Why glass correcion is +14D; +11.5D? Do we have any information about refraction?
- please consider moving laboratory findings and additional clinical information about patient from discussion section to case presentation
- line 59 - please correct "eyes" for 'eye'
- beside serous retinal detachment and uveal thickenning no information in both clinical findings and ultrasound examination about uveal effusion and detachment (only thickened uvea in ultrasound examination)
- influence of cataract surgery on UES was not discussed
- in discussion poor representation of newest publications about UES: e.g.Maggio et al. 2016, Claeys et. al 2020, Kaewsangthong et a. 2020
- decision, role and effect of of vitrectomy was not sufficentely discussed eg. Ohkita T. 2008
Author Response
Dear Reviewer 3,
Thank you for your review and your notes.
We tried to correct our manuscript according to your suggestions:
- No documentation or history of amblyopia was noted. (added to the text line - 58)We obtained only the refferal letter from his ophthalmologist. No other documentation are avaible.
2.laboratory findings are in case presentation. In discussion only some specific findings and their interpretation (HLA antigines) are discussed.
3." eyes" corrected
- cataract surgery influence added - line 164
- Maggio et al. 2016, Claeys et. al 2020 publications added, although they describe only indiopatic UES, only one patient in Maggio has nanophthalmic UES.
- Role of vitrectomy discussion added - line 192
We hope you will be satisfied with our revisions and you will consider our manuscript suitable for publication in journal Medicina.
Thank you for your kind response
Best regards

Round 2
Reviewer 2 Report
Dear Authors,
Although some changes have been made to the events order in the introduction and some little changes to the English language, I still think that these 2 things should be improved. Furthermore the real problem of this interesting case report is the lack of originality in terms of treatment option of UES.
Reviewer 3 Report
Thank You for changes of the manuscript.
The manuscript presents interesting and very rare material. The different treatment strategies with different results augment value of presented material.
However important role in presented material plays problems with diagnosis. To ensure readers and to enhance presentation of the case, if it is available, I would suggest providing better US scan exemplifying both retinal and uveal pathology, more representative for UES.